# In Vitro Antibiofilm Activity of Resveratrol against *Aeromonas hydrophila*

**DOI:** 10.3390/antibiotics12040686

**Published:** 2023-03-31

**Authors:** Ting Qin, Kai Chen, Bingwen Xi, Liangkun Pan, Jun Xie, Liushen Lu, Kai Liu

**Affiliations:** 1Key Laboratory of Aquatic Animal Nutrition and Health, Freshwater Fisheries Research Center, Chinese Academy of Fishery Science, Wuxi 214081, China; 2Wuxi Fisheries College, Nanjing Agricultural University, Wuxi 214081, China

**Keywords:** resveratrol, *Aeromonas hydrophila*, biofilm formation, motility, RNA-seq, quorum sensing

## Abstract

*Aeromonas hydrophila* is a Gram-negative bacterium that widely exists in various aquatic environments and causes septicemia in fish and humans. Resveratrol, a natural polyterpenoid product, has potential chemo-preventive and antibacterial properties. In this study, we investigated the effect of resveratrol on *A. hydrophila* biofilm formation and motility. The results demonstrated that resveratrol, at sub-MIC levels, can significantly inhibit the biofilm formation of *A. hydrophila*, and the biofilm was decreased with increasing concentrations. The motility assay showed that resveratrol could diminish the swimming and swarming motility of *A. hydrophila*. Transcriptome analyses (RNA-seq) showed that *A. hydrophila* treated with 50 and 100 μg/mL resveratrol, respectively, presented 230 and 308 differentially expressed genes (DEGs), including 90 or 130 upregulated genes and 130 or 178 downregulated genes. Among them, genes related to flagellar, type IV pilus and chemotaxis were significantly repressed. In addition, mRNA of virulence factors OmpA, extracellular proteases, lipases and T6SS were dramatically suppressed. Further analysis revealed that the major DEGs involved in flagellar assembly and bacterial chemotaxis pathways could be regulated by cyclic-di-guanosine monophosphate (c-di-GMP)- and LysR-Type transcriptional regulator (LTTR)-dependent quorum sensing (QS) systems. Overall, our results indicate that resveratrol can inhibit *A. hydrophila* biofilm formation by disturbing motility and QS systems, and can be used as a promising candidate drug against motile Aeromonad septicemia.

## 1. Introduction

*Aeromonas hydrophila* is a typically zoonotic pathogen with a worldwide distribution in aquatic environment, and has different pathogenic potential between aquatic animals and mammals [1,2]. It can not only cause motile aeromonad septicemia in fish, leading to severe economic losses in aquaculture, but also cause various diseases in humans, such as gastroenteritis, skin infections, meningitis, and necrotizing fasciitis [3,4]. Traditionally, antibiotics are widely used to treat *A. hydrophila* infections. However, the wide and frequent use of antibiotics in fish farming has led to the emergence of antibiotic resistance in pathogens and treatment fail ure [5,6]. This phenomenon threatens the healthy development of aquaculture and safety of aquatic products. Therefore, novel strategies are required to combat drug resistant *A. hydrophila*.

*A. hydrophila* possesses a variety of virulence factors, among which biofilm has been well characterized and considered one of the most critical virulence factors for establishing infections [7,8,9]. Biofilms are sessile microbial communities that attach to surfaces using a self-produced extracellular polymeric matrix [10]. The bacteria in biofilms become much more resistant to antibiotic treatments and host defense compared to their free-floating state [11]. Biofilm formation greatly enhances the survival of pathogens in hosts, and results in the persistence of bacterial infections. The beginning of biofilm formation is the attachment of planktonic cells to a surface [12], which makes motility the most important factor in the early phase of biofilm formation [13,14]. Moreover, biofilm formation is also coordinated by the quorum sensing (QS) system in pathogens [15,16]. Once the biofilm structure is disrupted, the antibiotic restores its efficacy and the host resolves the bacterial infections. Therefore, the antibiofilm strategy has been considered as a promising approach to overcome the bacterial infections [17]. 

Resveratrol (3,5,4′-Trihydroxystilbene) is a small natural plant polyphenol belonging to the stilbene family. The stilbene structure consists of precisely two aromatic rings and phenolic hydroxyl groups with double bonds (C6-C2-C6 carbon skeleton) [18]. Resveratrol is present in various plants, such as grapes, berries, peanuts, and is also found in high concentrations of red wine. It exists both in a *cis*- and *trans*-isomer, with the *trans*-isomer being the predominant form in plants and the most studied due to its greater availability and higher stability [18]. Resveratrol exhibits antioxidant, anti-inflammatory, immunomodulatory, cardioprotective, and neuroprotective effects, as well as potential therapeutic effects on diabetes, cancer, and Parkinson’s disease [18,19,20]. In addition, resveratrol has antibacterial, anti-viral, and antifungal activity [21]. As a natural product, resveratrol was demonstrated to be an effective antibiofilm and anti-QS agent for several bacterial pathogens [22,23,24,25]. The antibiofilm effect of resveratrol (0.02 mg/mL) on *Escherichia coli* is mediated by repressing the expression of genes related to curli production, motility, and AI-2 QS system [10]. In methicillin-resistant *Staphylococcus aureus* (MRSA), 150 μg/mL resveratrol inhibited biofilm formation by disturbing the expression of genes associated with QS, surface and capsular polysaccharides [24]. As for *Fusobacterium nucleatum*, resveratrol attenuated the biofilm formation by decreasing the expression of genes involved in QS and virulence [22]. Resveratrol may inhibit bacterial biofilm formation by disturbing the QS system. Furthermore, resveratrol (≥100 μg/mL) had no antibiofilm activity on *S. aureus* ATCC6538 [26,27]. The molecular mechanism of the antibiofilm effect of resveratrol may be different among various bacteria.

Our previous study showed that resveratrol, at sub-minimum inhibitory concentration (MIC), significantly inhibited *A. hydrophila* biofilm formation and hemolytic activity, and reduced the mortality of crucian carp challenged with *A. hydrophila* [28]. However, the action mechanism responsible for the antibiofilm activity of resveratrol is complex and has not been completely elucidated. In the present study, we investigated the biofilm inhibitory activity of resveratrol against *A. hydrophila*, mainly focusing on the biofilm formation, biofilm structure, and motility. Furthermore, RNA-seq was also performed to screen genes closely associated with resveratrol treatment and attempt to explore its possible mechanisms through functionally annotated and pathway-enriched analysis.

## 2. Results

### 2.1. Effects of Resveratrol on Biofilm Formation

Crystal violet staining results showed that 100 μg/mL resveratrol significantly inhibited biofilm formation compared to the control group, while 50 μg/mL resveratrol did not dramatically change the biofilm formed by *A. hydrophila* NJ-35 (Figure 1A). With increasing concentrations of resveratrol, the amount of biofilm reduced to 90.27 ± 0.11% and 49.34 ± 0.10% at 50 and 100 μg/mL resveratrol, respectively. In addition, both 50 and 100 μg/mL resveratrol decreased the exopolysaccharides (EPS) production (Figure 1B), and 100 μg/mL resveratrol noticeably repressed the total biofilm protein production of *A. hydrophila* NJ-35 (Figure 1C). The above three experimental results showed a similar trend; that is, the indicators of biofilm formation sequentially decreased following the increasing concentrations of resveratrol.

### 2.2. Effects of Resveratrol on Biofilm Structure

Microscopic observations matched the quantitative biofilm data well (Figure 2). In the absence of resveratrol, the biofilm formed by *A. hydrophila* NJ-35 had a uniform distribution with a dense coverage of the coverslip. When treated with resveratrol, the biofilm formed by *A. hydrophila* NJ-35 was sparser than that in the control group, and the biofilm structure became looser as the concentration of resveratrol increased.

### 2.3. Effects of Resveratrol on Motility

The effects of resveratrol on the swimming and swarming motilities of *A. hydrophila* NJ-35 were also examined. As shown in Figure 3, resveratrol at 50 μg/mL did not affect the swimming motility, whereas resveratrol at 100 μg/mL significantly reduced the swimming motility of *A. hydrophila* NJ-35 (*p* < 0.05). In contrast, both 50 and 100 μg/mL resveratrol noticeably decreased the swarming motility of *A. hydrophila* NJ-35 in a dose-dependent manner (*p* < 0.05). These results suggested that the inhibition of biofilm formation by resveratrol at 50 μg/mL was more closely related to swarming rather than swimming, while resveratrol at 100 μg/mL was involved in both swimming and swarming motilities.

### 2.4. Cytotoxicity of Resveratrol on J774A.1, with or without A. hydrophila NJ-35 Infection

The cytotoxic effect of resveratrol, at sub-MIC concentrations on J774A.1 cells, was detected by LDH activity. As shown in Figure 4, the cytotoxicity of 1% DMSO on J774A.1 cells was less than 1%, and there was no significant difference in cytotoxicity between the 50 μg/mL resveratrol and 1% DMSO. However, the cytotoxicity of resveratrol on J774A.1 cells remarkably increased when co-cultured with 100 μg/mL resveratrol. To directly examine the antibacterial activity, J774A.1 cells were exposed to *A. hydrophila* NJ-35, treated with or without resveratrol. Compared to the drug-free group (only *A. hydrophila* NJ-35 infection), 50 μg/mL resveratrol significantly decreased the LDH release of J774A.1 cells, but no significant difference was observed between the 100 μg/mL resveratrol and drug-free groups. The results indicate that resveratrol within 50 μg/mL is not cytotoxic to J774A.1 cells and has a good protective effect on the J774A.1 cells infected by *A. hydrophila* NJ-35.

### 2.5. Transcriptome Analysis of A. hydrophila NJ-35 Treated with Resveratrol

As demonstrated above, *A. hydrophila* NJ-35 showed reduced biofilm formation and motility in a dose-dependent manner when treated with resveratrol. In order to determine the molecular mechanism of these changes, RNA-seq was performed on *A. hydrophila* NJ-35 with 0, 50 and 100 μg/mL resveratrol (coded as Res 0, Res 50 and Res 100 groups). The differentially expressed genes (DEGs) were analyzed to infer the candidate genes related to biofilm and motility affected by resveratrol treatment, and to reveal the possible function of these differential genes and related molecular mechanisms.

#### 2.5.1. Screening and Functional Enrichment Analysis

Using the criteria of fold-change ≥ 2 and *p*-value < 0.05, we identified 230 (90 up- and 140 downregulated) and 308 DEGs (130 up- and 178 downregulated) in the presence of 50 and 100 μg/mL resveratrol compared to cultures grown without resveratrol, respectively, of which 120 DEGs were common in both datasets (Figure 5A). After comparing Res 100 and Res 50 groups, 242 DEGs (130 up- and 109 downregulated) were detected, of which 17 DEGs also occurred in Res 50 vs. Res 0, but not in Res 100 vs. Res 0; and 109 DEGs appeared in Res 100 vs. Res 0, but not in Res 50 vs. Res 0 (Figure 5B).

To gain insight into the functions of the DEGs that were altered by resveratrol treatment, all of the DEGs were mapped to terms in the GO and KEGG databases. GO analysis showed that among the up- and down-regulated genes of *A. hydrophila* NJ-35 in Res 50 and Res 100 groups, “metabolic process”, and “cellular process” in biological process, “cell”, “cell part”, and “membrane” in the cellular component, “catalytic activity”, “binding”, and “transporter activity” in molecular function were the major enriched groups (Figure 6). In addition, we noted that the number of downregulated genes distributed in the biological process of “response to stimulus” and “locomotion” was much higher than the number of upregulated genes. The top 5 KEGG pathways were as follows: “phenylalanine metabolism”, “tyrosine metabolism”, and “bacterial chemotaxis” in both Res 50 and Res 100 groups, “phenylalanine, tyrosine and tryptophan biosynthesis” and “histidine metabolism” in Res 50 group, “plant–pathogen interaction” and “selenocompound metabolism” in Res 100 group (Figure 7). It can be found that “bacterial chemotaxis” was the key enrichment pathway of the DEGs in both Res 50 and Res 100 groups, which indicated that the treatment of resveratrol had a significant impact on the motility of *A. hydrophila* NJ-35.

#### 2.5.2. Analysis of DEGs Related to Biofilm Formation and Motility

In this study, we focused our analysis on the major genes which were related to biofilm formation and motility. The KEGG classifications associated with “cell motility” and “cellular community prokaryotes” can be found in both Res 50 and Res 100 groups, which contained 12 and 2 DEGs in the Res 50 group, 13 and 5 DEGs in the Res 100 group, respectively (Appendix A). Notably, most of them were downregulated after resveratrol treatment. Not surprisingly, the expressions of the DEGs in “bacterial chemotaxis (ko02030)” and “flagellar assembly (ko02040)” pathways were almost repressed in resveratrol-treated groups. Interestingly, two genes associated with type IV pilus were downregulated, while the other two genes, encoding Flp family proteins, were upregulated. In addition, two GGDEF-domain-containing proteins (encoded by *U876_RS00255* and *U876_RS04630*), that are required for the production of the second messenger cyclic-di-guanosine monophosphate (c-di-GMP) and a helix-turn-helix transcriptional regulator (encoded by *U876_RS01620*) belonging to LysR-Type transcriptional regulators (LTTRs), were decreased. Besides the genes mentioned above, resveratrol also inhibited the expression of other virulence factors, including OmpA, T6SS, lipase and extracellular proteases (protease, elastase and collagenase) (Table 1).

### 2.6. Validation of Differentially Expressed Genes by qRT-PCR

A total of five genes were randomly selected to be measured the relative mRNA transcript levels using real-time quantitative PCR. As shown in the Figure 8, the qRT-PCR results showed similar expression tendency as the RNA-seq data, despite some quantitative differences at the expression level. The result suggested that the transcription abundance of DEGs in transcriptome analysis was highly reliable.

## 3. Discussion

*A. hydrophila*, one of the most common bacterial pathogens in aquaculture environments, has been reported to be resistant to a number of antibiotics [29,30]. The role of biofilm formation in *A. hydrophila* pathogenesis is well established, as it provides the bacterium with enhanced tolerance to antimicrobial agents and host defenses [8,9,31]. Therefore, inhibiting biofilm formation is of great significance to combat the infection of *A. hydrophila*. Antibiotic therapy typically alleviates the symptoms caused by planktonic bacteria, but fails to kill bacteria in biofilms [32]. During aggravation of bacterial resistance, plant-derived compounds have attracted much attention because of their safety, availability and low toxicity. Resveratrol is a natural plant polyphenol that occurs in various plants, and has been demonstrated to have antibiofilm effects on many Gram-negative and Gram-positive bacterial pathogens [33]. In the present study, resveratrol, at the sub-inhibitory concentrations, can significantly inhibit the biofilm formation of *A. hydrophila* NJ-35 in a dose-dependent manner, which was consistent with our previous findings [28]. Thus, we further investigated the action mechanism of resveratrol on *A. hydrophila* biofilms.

Bacterial adhesion and colonization play an important role in the process of biofilm formation [34]. With the formation of biofilm, bacterial pathogenicity enhances significantly. In our study, the biofilms formed by *A. hydrophila* NJ-35 on the slides were destroyed by resveratrol; meanwhile, as the resveratrol concentration increased, the biofilms decreased gradually and became sparsely distributed on the slides, which had a similar trend to the results obtained by crystal violet staining, EPS production and the total biofilm protein.Therefore, we speculated that resveratrol decreased the adhesion of *A. hydrophila* NJ-35, thus inhibiting biofilm formation. This finding was also confirmed in *S. aureus* and avian pathogenic *E. coli* [24,35].

Motility has a positive influence on the development of biofilm formation, as it facilitates pathogens to colonize, adhere and invade the host cells [36,37]. *Aeromonas* species possess polar flagella for swimming motility and lateral flagella for swarming motility over surfaces [38]. The main antibiofilm activities of resveratrol comprised the inhibition of QS and motility [21]. Our study showed that resveratrol diminished the swimming and swarming motility of *A. hydrophila* NJ-35 in a dose-dependent manner. Particularly, resveratrol dramatically repressed swarming motility even at 50 μg/mL. In EHEC, *trans*-resveratrol inhibited both swimming and swarming motility, and suppressed the expression of several key motility and flagellar genes, including *flhD*, *fimA*, *fimH*, and *motB* [10]. Additionally, reduced swarming ability has also been reported for *Proteus mirabilis* and *V. vulnificus* [39,40]. Given the above, resveratrol might impel the biofilm formation of *A. hydrophila* NJ-35 via disrupting its adhesion and inhibiting swimming and swarming motility.

We further explored the molecular mechanism of these changes through transcriptome analyses (Figure 9). With the increasing concentration of resveratrol, numbers of DEGs in the Res 50 and Res 100 groups increased. As was expected, a large number of DEGs were significantly enriched in biofilm- and motility-related pathways, especially in flagellar and type IV pilus assembly, as well as in bacterial chemotaxis. It has been reported that flagellar motility and chemotaxis exert profound influences on bacterial behaviors, including swarming, biofilm formation and auto-aggregation; the type IV pilus plays an important role in structural biofilm development, including initial formation and dispersal [41,42]. Thus, Resveratrol indeed regulated the biofilm formation and motility of *A. hydrophila* at the transcriptional level.

It is known that the GGDEF-domain-containing proteins contribute to the synthesis of the intracellular signaling molecule c-di-GMP. The latter coordinates the bacterial lifestyle transition from motility to sessility and vice versa [43]. Generally, low c-di-GMP levels are conducive to bacterial motility, such as swimming, twitching and swarming, and inconducive to biofilm formation. In contrast, high c-di-GMP levels promote biofilm formation and restrict bacterial motility [43,44]. Kozlova et al. [45] have demonstrated that c-di-GMP overproduction in *A. hydrophila* SSU dramatically increased the transcripts of *luxS*, *litR*, *vpsT*, *fleQ*, and *fleN*, thus enhancing biofilm formation, but reducing motility, which was consistent with that of the Δ*luxS*. Unlike Δ*luxS*, overproduction of c-di-GMP in Δ*ahyRI* resulted in a slight increase in biofilm formation with no effect on motility, due to the high level of c-di-GMP upregulating the transcriptional level of *vpsR* and downregulating the levels of *vpsT* and *fleN*. On the contrary, Δ*qseB* exhibited a significant reduction in biofilm formation with no effect on swimming motility when c-di-GMP was overproduced, which was correlated with altered levels of *fleN* and *vpsT* [46]. In addition, resveratrol has been demonstrated to repress the expression of *ahyR* and *ahyI*, as well as T6SS [15,47]. Thus, we indicated that resveratrol may inhibit *A. hydrophila* biofilm formation though AhyRI QS system in a c-di-GMP manner. However, further studies are necessary to demonstrate this.

The LTTRs are the largest family of diverse and well-characterized global transcriptional regulators of prokaryotes [48]. This family of proteins is involved in the regulation of various processes, including multidrug resistance, virulence, QS, motility and biofilm formation [49,50,51]. An in-frame deletion of *Bcal3178* (a LysR-type regulator) caused a significant downregulation of biofilm formation and protease production, which are controlled by QS systems in *Burkholderia cenocepacia* [50]. Four putative LTTR family proteins (A0KIU1, A0KJ82, A0KPK0, and A0KQ63) were decreased in *A. hydrophila* following antibiotic treatment, and the deletion of A0KQ63 exhibited multidrug resistance properties [52]. In this study, U876_RS01620 encoding a LTTR was downregulated after adding resveratrol in *A. hydrophila*, while the expression of the two other genes encoding the SMR family of multidrug efflux pumps were repressed. One possible explanation is that the protein encoded by *U876_RS01620* may not affect the above two genes, and resveratrol may activate other pathways to regulate SMR family proteins. In this regard, it may be of interest to evaluate whether resveratrol regulates drug resistance of *A. hydrophila* through the LTTR and whether the LTTR is mediated by the c-di-GMP. 

The pathogenesis of *A. hydrophila* is multifactorial, and characterized by the involvement of a number of virulence factors, such as adhesins, outer membrane proteins, hemolysins, protease, and secretion system [53,54]. Our study showed that resveratrol dramatically decreased the gene expression of OmpA, lipase, protease, elastase, collagenase, and T6SS. In addition, resveratrol has been demonstrated to affect the QS-related gene expression, weaken the hemolytic activity in vitro, and attenuate the in vivo virulence of *A. hydrophila* in the crucian carp infection, supporting the protective role of resveratrol against fish disease [28,47]. Following these results, we speculated that the natural compound resveratrol might inhibit *A. hydrophila* biofilm formation by disturbing the c-di-GMP- and LTTR-dependent QS systems, which could regulate adhesion, motility and virulence. Further study needs to be performed.

## 4. Materials and Methods

### 4.1. Bacterial Strains and Growth Conditions

*A. hydrophila* NJ-35 (CGMCC No.8319) was cultured in Luria broth (LB) or on LB agar at 28 °C [55]. Resveratrol (99%, Aladdin, Shanghai, China) was dissolved in dimethyl sulfoxide (DMSO) as a 10 mg/mL stock solution, and diluted to the required working concentrations depending on the assay type.

### 4.2. Crystal Violet Biofilm Assay

An assay of static biofilm formation was performed in 96-well polystyrene plates, as previously reported [56]. An overnight culture of *A. hydrophila* NJ-35 was collected and normalized to 1 × 10^7^ CFU/mL, then diluted 1:100 in a fresh LB medium. In our earlier studies, the MIC of resveratrol against *A. hydrophila* NJ-35 was recorded as 1024 μg/mL, whereas the resveratrol concentration below 64 μg/mL did not affect the growth of *A. hydrophila* NJ-35 [28]. Resveratrol was added for experimental cultures at final concentrations of 50 and 100 μg/mL, respectively. DMSO (0.1%, *v*/*v*) was added as the control group. Two hundred microliters of the above dilutions were dispensed to the wells of microtiter plates and incubated at 28 °C for 48 h without agitation. Following incubation, cell growth was measured at 600 nm. Then, the suspended culture was poured out and the wells were washed three times with sterile phosphate-buffered saline (PBS). The adherent cells were fixed with 200 μL methanol for 15 min and dried at room temperature. Subsequently, each well was stained with 200 μL of 1% (wt/vol) crystal violet solution for 15 min and washed with PBS to remove the unbound dye. The formed biofilm was dissolved in absolute ethanol, and the absorbance was measured at 590 nm using a spectrophotometer (MultiskanGO, Thermo Scientific, Vantaa, Finland). The results were normalized to reduce the differences caused by bacterial growth rates according to the method of Niu and Gilbert [57].

### 4.3. Exopolysaccharides Assay

*A. hydrophila* NJ-35 was cultured with or without resveratrol in 6-well plates, at 28 °C for 48 h. The formed biofilms were washed three times with PBS, then resuspended in 0.85% NaCl containing 0.22% formaldehyde. After centrifugation, the supernatants were collected and the EPS were determined by the phenol-sulfuric acid method [58].

### 4.4. Total Biofilm Protein Assay

The biofilms formed in 6-well plates were resuspended in 1 mL PBS. After ultrasonication, the total biofilm protein concentrations were detected using the Modified Bradford Protein Assay Kit (Sangon Biotech, Shanghai, China).

### 4.5. Scanning Electron Microscopy

The inhibitory effect of resveratrol on the biofilm formation of *A. hydrophila* NJ-35 was observed using SEM. Overnight bacterial culture was normalized to 1 × 10^7^ CFU/mL and diluted 1:100 in LB medium with 50 and 100 μg/mL resveratrol, respectively, and 0.1% (*v*/*v*) DMSO was added as a control. Pre-sterilized coverslips (Φ = 14 mm) were placed into 12-well plates, then 1 mL bacterial dilution was dispensed to each well and incubated at 28 °C for 48 h. The slides were rinsed well with PBS and fixed with 2.5% glutaraldehyde for 4 h. The samples were dehydrated with a series of gradient acetone (10, 30, 50, 70, 90, and 100%, *v*/*v*) for 15 min, and dehydrated in 100% acetone twice. Finally, the sample was observed under a scanning electron microscope (S-4800, Hitachi, Tokyo, Japan). Nine random positions in three independent experiments were chosen for microscopic analysis.

### 4.6. Lactate Dehydrogenase Assay

J774A.1 murine macrophages were cultured in DMEM medium with high glucose (HyClone, Beijing, China) containing 10% fetal bovine serum (FBS, Every green, Hangzhou, China) at 37 °C with 5% CO_2_. Cells (1 × 10^5^ cells/well) were cultured in 96-well plates for 24 h, then washed three times with PBS. Cells were divided into two groups: resveratrol treatment group and *A. hydrophila* NJ-35 + resveratrol treatment group. DMEM medium and 1% DMSO served as negative control and solvent control, respectively. For infection, bacteria grown to logarithmic phase were collected, washed and seeded with or without resveratrol into each well at the multiplicity of infection (MOI) of 1:1. After 3 h of incubation, the supernatant was harvested and the LDH activity was measured by the LDH Cytotoxicity Assay Kit (Invitrogen, Carlsbad, CA, USA).

### 4.7. Swimming and Swarming Motility

An LB medium with 0.3% agar for swimming motility and 0.5% agar for swarming motility were prepared, as described previously. Resveratrol was added to the LB agar to final concentrations of 50 or 100 μg/mL. Additionally, DMSO (0.1%) was added as a control. Overnight bacterial culture was normalized to 1 × 10^7^ CFU/mL with the LB medium. Then, 5 μL culture was spotted into the middle of the plate and incubated at 28 °C for 24 h. Motility was assessed by measuring the migration diameter of bacteria from the inoculation point to the periphery of the plate.

### 4.8. Transcriptome Analysis

*A. hydrophila* NJ-35 was inoculated into 100 mL LB medium, supplemented with 50 and 100 μg/mL resveratrol at an initial OD_600_ of 0.05, and cultured at 28 °C until the OD_600_ reached 0.8. Bacteria treated with 0.1% DMSO were set as a control group. Cells were collected and washed with PBS for RNA-seq using Illumina HiSeq^TM^ 2500 at Shanghai OE Biotech Co., Ltd. (Shanghai, China) Three parallel samples for each group were pooled as biological replicates for transcriptome analyses. Reads were aligned to the reference genome NZ_CP006870.1 [55] using Rockhooper2 [59]. Gene transcript expression levels were calculated by RPKM [60]. Differential expression analysis was conducted using DESeq [61], then the DEGs picked out, such that *p*-value < 0.05 and difference of multiples >2. GO and KEGG enrichment analyses of DEGs were performed by hypergeometric distribution tests to determine the biological functions or pathways that are mainly affected by differential genes.

### 4.9. Real-Time Quantitative PCR (qRT-PCR) Verification

Total RNA was extracted from bacteria with RNAiso Plus (TaKaRa, Tokyo, Japan), according to the manufacturer’s instructions, and quantified to 40 ng/µL with a Nanodrop 2000 (Thermo Fisher Scientific, Wilmington, MA, USA). Two microliters of diluted RNA were directly used for qRT-PCR with One Step SYBR^®^ PrimeScript^®^ Plus RT-PCR Kit (TaKaRa, Dalian, China). The primers were listed in Table 2. The expression levels of the tested genes were analyzed with *rpoB* as the reference gene. Fold-change was calculated using the 2^−ΔΔCt^ method [62].

### 4.10. Statistical Analysis

All experiments were repeated independently three times. The data were expressed as the mean ± SD. A one-way analysis of variance (ANOVA) was conducted to identify the significant differences, followed by Bonferroni’s post-test (IBM SPSS Statistics, version 19.0, Armonk, NY, USA). A *p*-value of <0.05 or <0.01 was considered statistically significant. 

## 5. Conclusions

Our study highlights that resveratrol at sub-MIC has an inhibitory effect on the biofilm formation and motility of *A. hydrophila*. Transcriptome analysis found that resveratrol significantly repressed bacterial chemotaxis and flagellar assembly pathways, disrupted type IV pilus synthesis, downregulated the c-di-GMP and LTTR levels, which all involved in QS systems. Thus, we concluded that resveratrol could decrease biofilm formation at concentrations without anti-*A. hydrophila* growth by inhibiting QS systems. Additionally, resveratrol also markedly suppressed the gene expression of several important virulence factors, such as OmpA, extracellular proteases, lipases, and T6SS. In conclusion, resveratrol could be considered a potential therapeutic drug by attenuating the capacity of pathogenic *A. hydrophila* to cause infection, and is unable to induce muti-drug resistance.

## Figures and Tables

**Figure 1 antibiotics-12-00686-f001:**
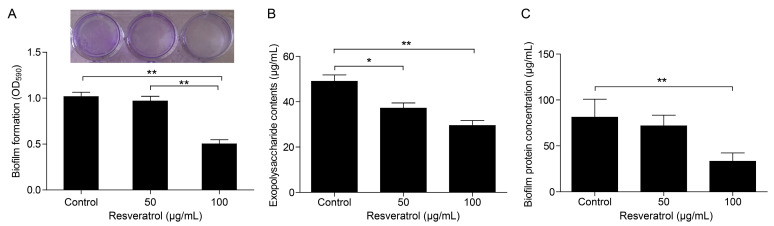
Inhibitory effects of resveratrol on biofilm formation of *A. hydrophila* NJ-35. (**A**) Crystal violet staining assay. (**B**) Exopolysaccharide production of formed biofilm. (**C**) The total protein of biofilm. The data were presented as the mean ± SD (n = 3) of three independent experiments. * *p* < 0.05, ** *p* < 0.01.

**Figure 2 antibiotics-12-00686-f002:**
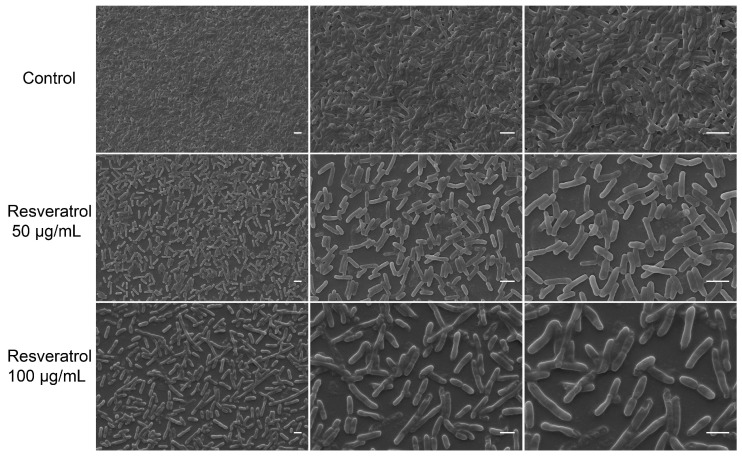
Scanning electron micrographs (SEM) of biofilm formed by *A. hydrophila* with different concentrations of resveratrol. Bars, 2 μm at magnification 2000 (**left**), 4000 (**medial**), and 6000 (**right**).

**Figure 3 antibiotics-12-00686-f003:**
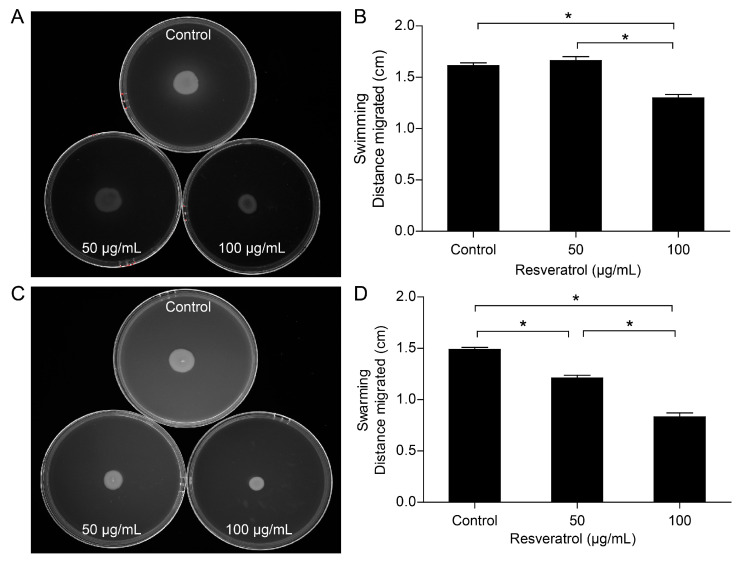
Effects of resveratrol on the swimming and swarming motility of *A. hydrophila*. The abilities of swimming (**A**,**B**) and swarming (**C**,**D**) motility were assessed by examining the migration of bacteria through the agar, from the center toward the periphery of the plate. The results were reproduced in three independent experiments, and the error bars represent SDs. * *p* < 0.05.

**Figure 4 antibiotics-12-00686-f004:**
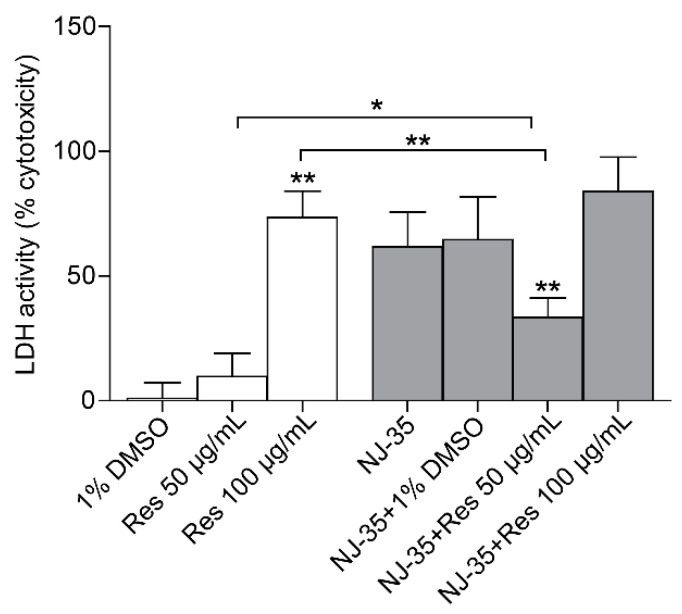
LDH release of J774A.1 cells with or without *A. hydrophila* NJ-35 treatment, and the indicated concentrations of resveratrol. The data were presented as the mean ± SD (n = 6). * *p* < 0.05, ** *p* < 0.01.

**Figure 5 antibiotics-12-00686-f005:**
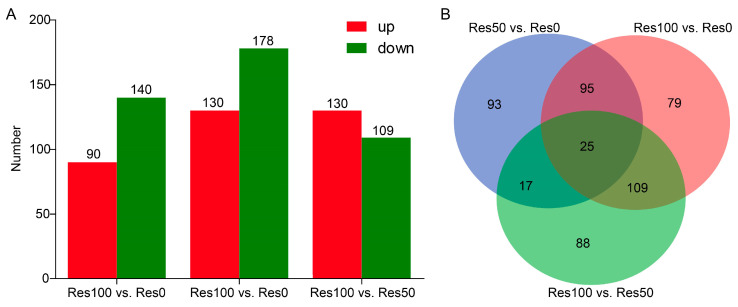
Differentially expressed gene (DEG) analysis. (**A**) Number of upregulated and downregulated DEGs between resveratrol-treated and control groups. (**B**) Venn analysis between resveratrol-treated and control groups.

**Figure 6 antibiotics-12-00686-f006:**
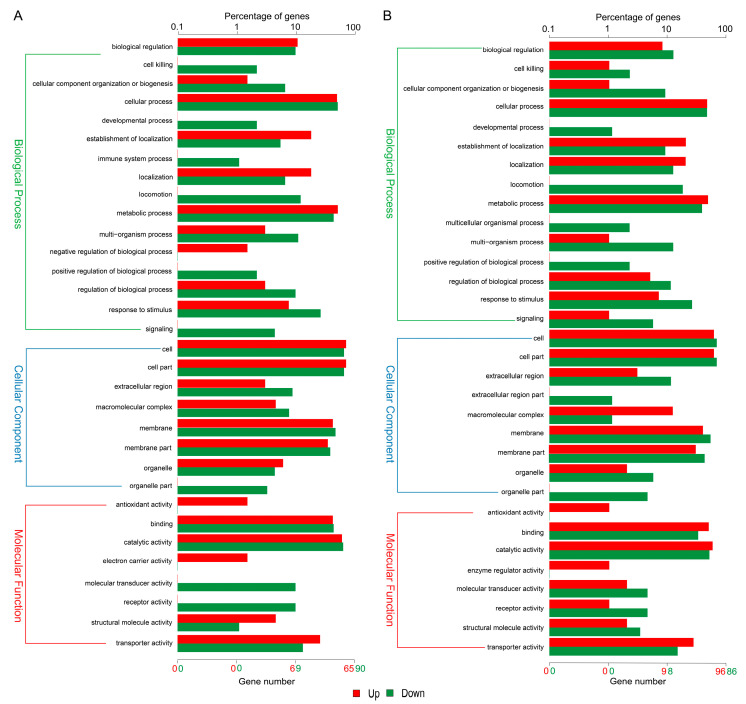
GO functional annotation for DEGs. (**A**) Res 50 vs. Res 0 groups. (**B**) Res 100 vs. Res 0 groups.

**Figure 7 antibiotics-12-00686-f007:**
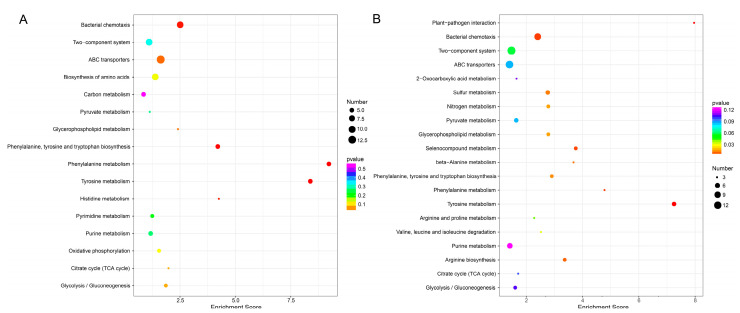
Top 20 DEGs of KEGG pathway enrichment. (**A**) Res 50 vs. Res 0 groups. (**B**) Res 100 vs. Res 0 groups.

**Figure 8 antibiotics-12-00686-f008:**
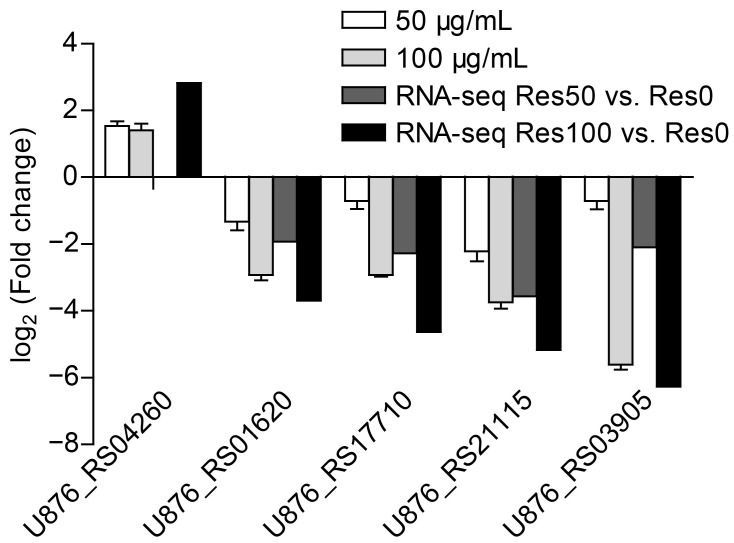
Validation of four different randomly selected genes by qRT-PCR.

**Figure 9 antibiotics-12-00686-f009:**
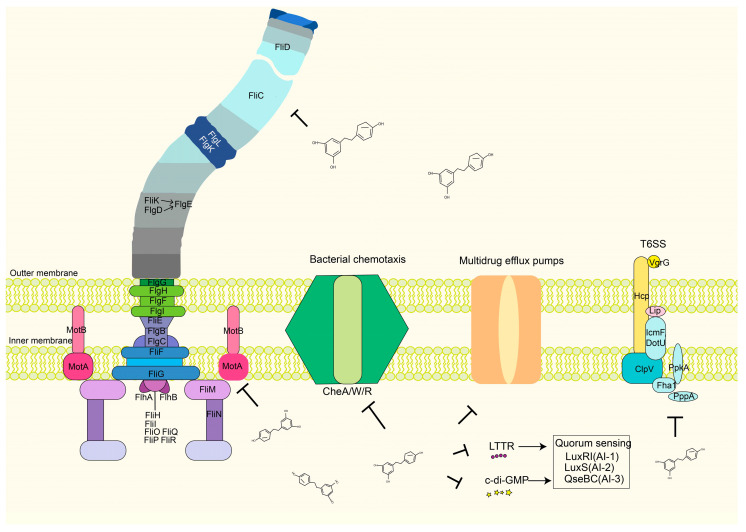
Schematic diagram of the inhibitory effects of resveratrol on *A. hydrophila*. “⊤” denotes inhibition.

**Table 1 antibiotics-12-00686-t001:** A list of some of the important differentially expressed genes between the resveratrol-treated and untreated *A. hydrophila* NJ-35, including genes related to motility, biofilm formation and other virulence factors.

Gene ID	log2 Fold-Change		Description
	Res 50/Res 0	Res 100/Res 0	
Motility			
U876_RS13815	−1.55	−3.00	Flagellin
U876_RS13830	-	−3.12	Flagellin
U876_RS13835	-	−2.50	Flagellin-like protein
U876_RS15700	−1.80	-	Flagellar biosynthesis protein FlhB
U876_RS05370	−1.39	-	Type IV pilus biogenesis protein PilO
U876_RS15085	1.89	-	Flp pilus assembly protein CpaB
U876_RS15095	2.30	2.04	Flp family type IVb pilin
U876_RS19675	−1.45	−2.03	type IV pilin
Chemotaxis			
U876_RS17720	−2.20	−3.57	Chemotaxis protein CheA
U876_RS17730	−2.77	−2.89	Chemotaxis protein CheW
U876_RS17740	−1.96	−3.06	Protein-glutamate O-methyltransferase CheR
U876_RS21115	−3.57	−4.71	Chemotaxis protein
c-di-GMP			
U876_RS00255	-	−1.81	GGDEF-domain-containing protein
U876_RS04630	−4.55	−5.06	GGDEF-domain-containing protein
Extracellular Proteases			
U876_RS04035	−1.86	−2.85	Protease
U876_RS18875	−2.76	−4.74	Elastase
U876_RS20565	−2.94	−3.58	Collagenase
Lipase			
U876_RS20585	−1.66	−2.34	Lipase
U876_RS20590	−1.68	−1.93	Lipase chaperone
T6SS			
U876_RS21275	−1.47	-	Type II/IV secretion system protein
U876_RS13095	−1.72	-	Type VI secretion system baseplate subunit TssK
U876_RS13100	−1.73	-	Type VI secretion system lipoprotein TssJ
U876_RS13105	−1.69	-	Type VI secretion system-associated FHA domain protein TagH
U876_RS13110	−1.34	-	Type VI secretion system baseplate subunit TssG
U876_RS13155	-	−1.81	Type VI secretion system tip protein VgrG
Small drug resistance			
U876_RS03760	−3.43	-	SMR family multidrug efflux pump
U876_RS03765	−2.91	−2.21	SMR family multidrug efflux pump
Others			
U876_RS01620	−1.93	−3.42	Helix-turn-helix transcriptional regulator
U876_RS08385	-	−2.26	OmpA family protein

**Table 2 antibiotics-12-00686-t002:** Primers used in qRT-PCR.

Gene ID	Description	Primer	Sequence (5′ to 3′)	Reference
U876_RS04260	Universal stress protein	F	CCACAAGGCTGAACTCAA	This study
		R	CAGGTCGGCTTTCTCTTC	
U876_RS17710	Response regulator	F	CGGTTATGAGGTGATGGAG	This study
		R	TTCCTGCTTCTTGCTGTC	
U876_RS21115	Chemotaxis protein	F	TGCTGTACGCCTTCTAATG	This study
		R	CATGCTGTAGTGCTGACC	
U876_RS03905	Competence protein ComEA	F	ATGAACTACAAGACCCTGAC	This study
		R	GATCCACGGTAGTGAACTT	
U876_RS01620	Helix-turn-helix transcriptional regulator	F	GCGATCTGGTCAACTACTA	This study
		R	GCGGTTCTTCACATTCAAT	
*ropB*	Housekeeping gene, RNA polymerase beta subunit	F	ACCGACGAAGTGGACTATCT	[53]
		R	CGGCGTTCATAAAGGTGGAT	

## Data Availability

Data are available upon request.

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
