# Peer review of "In Vitro Antibiofilm Activity of Resveratrol against Aeromonas hydrophila"

_antibiotics, 2023, doi:10.3390/antibiotics12040686_

Round 1

Reviewer 1 Report

Qin et. al., showed the effect of resveratrol against Aeromonas hydrophila mediated biofilm formation. This is a nice work; I would suggest adding few things to strengthen the work.

1. The experiment missing the positive control. I would suggest using commercially available antibiotic used against Aeromonas hydrophila to compare the effect of resveratrol.

2. I would suggest quantifying the total biofilm protein concentration with or without treatment.

3. I would suggest seeing the cytotoxic effect of those sub-MIC concentrations in mammalian cells.

Reviewer 2 Report

This article is well written. Some comments to be addressed in this article are:

1. Abstract has presented Introduction, Objective, methods, and results. However, the short conclusion is missing. Please provide this in the last abstract.

2. The introduction is interesting. The authors have highlighted the novelty of this study in regard to mechanism of resveratrol as anti-biofilm. The authors are suggested to add the chemical structure of Resveratrol.

3. In Introduction, the gap analysis must be more explored in introduction.

4. The symbol %, should be without space; 90.27 % should be 90.27%

5. The amount of biofilm decreased is suggested to be expressed by mean and SD

6. The mechanism underlying the mechanism of resveratrol should be highlighted.

7. Are the primers used from author's design or from references. Please state

Round 2

Reviewer 1 Report

Thank you for addressing my concerns and I would be happy to recommend the work for publication.